# Systematic Review and Meta-Analysis of Oral Anticoagulant Therapy in Atrial Fibrillation Cancer Patients

**DOI:** 10.3390/cancers15092574

**Published:** 2023-04-30

**Authors:** Alberto Cereda, Stefano Lucreziotti, Antonio Gabriele Franchina, Alessandra Laricchia, Valentina De Regibus, Barbara Conconi, Matteo Carlà, Andrea Spangaro, Matteo Rocchetti, Luca Ponti, Alessandro Minardi, Elena Sala, Giuseppe Massimo Sangiorgi, Gabriele Tumminello, Lucia Barbieri, Stefano Carugo, Paolo Aseni

**Affiliations:** 1Cardiovascular Department, Association Socio Sanitary Territorial Santi Paolo e Carlo, 20153 Milano, Italy; 2Division of Cardiology, “Tor Vergata” University Hospital, 00133 Rome, Italy; 3Department of Biomedicine and Prevention, “Tor Vergata” University of Rome, 00133 Rome, Italy; 4Cardiology Unit, Fondazione IRCCS Ca’ Granda Ospedale Maggiore Policlinico, Department of Clinical Science and Community Health, University of Milan, 20122 Milan, Italy; 5Department of Emergency, ASST Grande Ospedale Metropolitano Niguarda, 20162 Milan, Italy; 6Department of Biomedical and Clinical Sciences “L. Sacco”, Università degli Studi di Milano, 20157 Milan, Italy

**Keywords:** atrial fibrillation in cancer patients, anticoagulants, cardio-oncological prevention

## Abstract

**Simple Summary:**

Cancer and atrial fibrillation share an enhanced bleeding and thrombotic risk, nevertheless in cancer populations optimal anticoagulation regiments are yet to be validated. This meta-analysis aimed to gather the currently available evidence addressing the use of DOACs in atrial fibrillation cancer patients compared to Warfarin. In the oncological population, DOACs confer benefits in terms of stroke, major and minor bleeding, and thrombotic events, although prospective studies are awaited to strengthen the body of evidence currently available.

**Abstract:**

(1) Introduction: Cancer and atrial fibrillation (AF) are increasingly coexisting medical challenges. These two conditions share an increased thrombotic and bleeding risk. Although optimal regimens of the most suitable anti-thrombotic therapy are now affirmed in the general population, cancer patients are still particularly understudied on the matter; (2) Aims And Methodology: This metanalysis (11 studies (incl. 266,865 patients)) aims at evaluating the ischemic-hemorrhagic risk profile of oncologic patients with AF treated with oral anticoagulants (vitamin K antagonists vs. direct oral anticoagulants); (3) Results: In the oncological population, DOACs confer a benefit in terms of the reduction in ischemic, hemorrhagic and venous thromboembolic events. However, ischemic prevention has a non-insignificant bleeding risk, lower than Warfarin but significant and higher than the non-oncological patients; (4) Conclusions: Anticoagulation with DOACs provides a higher safety profile with respect to VKAs in terms of stroke reduction and a relative bleeding reduction risk. Further studies are needed to better assess the optimal anticoagulation strategy in cancer patients with AF.

## 1. Introduction

The coexistence of cancer and atrial fibrillation (AF) is an increasingly reported medical issue due to longer life expectancy in the general population along with the higher survival rate of oncological patients. Therapies for these clinical entities are nevertheless constantly improving: better anti-neoplastic strategies for the former, safer medications (antiarrhythmics and anticoagulants), and interventional procedures (ablation and percutaneous closure of the left atrial appendage) for the latter [1].

Current reports have demonstrated that newly diagnosed cancer patients have a 47% higher risk of developing AF when compared to noncancer patients. More specifically, when considering two of the most frequently diagnosed cancer types, colorectal and breast, the risk of suffering from AF is 54% and two times higher respectively when compared to noncancer patients.

Alterations in hemostasis and inflammation represent part of the common pathophysiological ground linking these pathologies. A consistent body of literature has linked AF to a pro-thrombotic state. Similarly, cancer patients experience a higher risk of thrombotic manifestations but are concomitantly exposed to a superior bleeding risk compared to non-oncological populations [2,3].

Cancer patients share several factors predisposing to AF development such as direct cancer effect (i.e., proximity of masses to the heart), paraneoplastic manifestations, inflammation, medical therapies (cytotoxic chemotherapy; high doses of steroids and other), surgical therapy, cancer-related comorbidities and autonomous nervous system imbalance (mostly due to pain, physical and emotional stress).

Furthermore, cancer increases the risk of suffering from AF consequences such as the development of thrombi and cardioembolic stroke. The procoagulant state is a well-known and largely studied feature of cancer.

Bleeding is also a frequent problem for patients with advanced cancer posing, in some cases, life-threatening risks. Bleeding is caused by cancer itself due to neo-angiogenesis, vessel invasion, or tumor regression. Bleedings may also be exacerbated by anti-tumor treatments including intravenous and oral therapies, radiation therapy, and surgery. Thrombocytopenia, which is a common feature for both solid and nonsolid malignancies, can also predispose to bleeding.

In AF, anticoagulation and its management have been extensively studied and optimized both for medication choice and therapy duration. In addition to the more classic heparins and vitamin K antagonists (VKAs), a relatively new option for the treatment and prevention of AF-related cardio-embolic events is represented by direct oral anticoagulants (DOACs). These include direct thrombin inhibitors (Dabigatran) and Factor Xa inhibitors (Rivaroxaban, Apixaban, and Edoxaban). DOACs are particularly attractive because of their oral administration—avoiding the prolonged use of parenteral drugs such as heparins—along with fixed doses, a lower rate of food and medications interactions, and no need for activity monitoring, all major pitfalls of VKA therapy [4,5,6,7].

Despite an altered coagulation profile, the oncologic population is largely understudied in the field of anti-thrombotic therapies, and several reasons can be identified for the shortage of literature on the matter. Firstly, the most common risk scores to assess ischemic and hemorrhagic profiles (CHA2DS2-VASc and HAS-BLED scores, respectively) do not consider cancer among their variables. Secondly, the oncological ischemic-hemorrhagic risk profile differs according to several variables, including the stage of the disease, the type of cancer, and the underlying cardiovascular profile [8].

As cancer patients were not included in large cardiovascular studies, the potential benefit of direct oral anticoagulants (DOACs) appears to be reasonable but not supported by current evidence [9,10,11].

## 2. Purpose of the Study

The purpose of the study is to confirm the benefit of DOACs in cancer patients and evaluate therapy efficacy in terms of stroke reduction compared to major bleeding events.

## 3. Study Design

This review was registered in the International Prospective Register of Systematic Reviews (PROSPERO) and was conducted and presented according to best practice recommendations, including the Preferred Reporting Items for Systematic Reviews and Meta-Analyses (PRISMA) reporting guidelines [12].

## 4. Search Strategy

An online search was conducted using the MEDLINE database. Searches were conducted by authors AS and MR and were updated to February 2023. We used the following key concept: “cancer”, “atrial fibrillation” and “anticoagulation”. To build our search strategy appropriate text words and MeSH terms were developed. From each article, further references were identified and included in the research according to backward snowballing. The search strategy is reported according to PRISMA methodology in Appendix A.

## 5. Study Selection

The studies were evaluated by the two reviewers MR and AS. Considering the limited evidence on the subject, non-randomized clinical studies and retrospective registries were deliberately included in the metanalysis as well. Inclusion criteria comprised articles in the English language, on human subjects, reporting the odds ratios (ORs) or hazard ratios (HRs) and the corresponding 95% confidence intervals (CIs) and comparing oral anticoagulant and warfarin in patients suffering from cancer and atrial fibrillation. From our search, we screened 719 studies, and 67 reports were selected based on the title of the article. After reading the abstract and the full text, 11 studies were found to be eligible for our analysis. All disagreements in the selections of the study were addressed by a third reviewer (AC). Studies included are reported and described in Table 1 and Appendix A.

## 6. Outcome and Endpoints

The following meta-analytical variables were considered: ischemic stroke, systemic embolism, venous thromboembolism, major bleeding, minor bleeding, gastrointestinal bleeding, cerebral bleeding, and overall mortality in the 1-year follow-up.

Covariates identified and subsequently used in the meta-regression were age, percentage of the female sex in the study, average CHA2DS2-VASc and HAS-BLED scores, type of cancer, and type of anticoagulant.

The “major bleeding” variable was reported according to different criteria in the various studies; therefore, in the present metanalysis, an arbitrary inclusive definition of major bleeding as “bleeding that involves sequelae or consumes major healthcare resources” was adopted. In equivocal cases, the discrepancy between minor and major bleeding was discriminated by reviewers AS and MR.

The number needed to treat (for ischemic events) and the number needed to harm (for bleeding events) were calculated using the Formula (1) divided by the absolute risk reduction number.

## 7. Bias Assessment

Table 1, Appendix A show the characteristics of the included studies. Appendix A shows the risk of bias divided into randomized (A) and observational (B) studies.

## 8. Statistical Analysis

Quantitative data are reported as mean values with standard deviation. Categorical variables are expressed in terms of percentages. A random effects model according to the DerSimonian and Laird method was used for meta-analysis. Results of the metanalysis are expressed in odds ratio (OR) with a 95% confidence interval. A random-effect metaregression for the covariates sex, age, HAS-BLED, and CHA2DS2-VASc was performed for statistically significant OR values. Funnel plots were visually inspected for the potential presence of small study effects, with regression tests to further corroborate the appraisal of publication bias (Appendix A). Statistical significance was set at the two-tailed 0.05 level. Computations were performed with R Software packages OpenMetanalyst and JASP.

## 9. Results

The ischemic and hemorrhagic risk profiles in cancer patients are equally increased, as reported in the cumulative meta-analysis of Figure 1. The risk of stroke is 5.2% (2.7–7.7) against a major bleeding event risk of 5.2% (4.2–6.2). Appendix A show that there is no potential selection bias as patients on DOAC therapy have non-different CHA2DS2-VASc and HAS-BLED scores compared to patients on Warfarin.

Although the ischemic and bleeding risks can be estimated from the CHA2DS2-VASc and HAS-BLED scores, our analysis found no significant correlation between events (strokes and major bleedings) and score points in the included studies (Appendix A).

The therapeutic strategy DOACs versus Warfarin determines a reduction in the incidence of ischemic stroke of 27% (OR 0.73 CI 95% 0.52–0.94; Figure 2 and Appendix A). Furthermore, an even more important reduction in major bleeding events to 42% (OR 0.58 95% CI 0.45–0.72; Figure 3 and Appendix A) can be observed. However, benefits are not only limited to these two covariates (stroke and bleeding), a reduction trend in systemic embolisms, even if non-significant, can be shown. (Figure 4, Appendix A). There is on the contrary a significant reduction in venous thromboembolism (Figure 4). The prevention of bleeding also comprises minor bleeding (OR 0.63 95% CI 0.40–0.86; Figure 5 and Appendix A). Concerning the one-year follow-up, there is no benefit on overall mortality in this cardio-oncology population (OR 0.86 95% CI 0.57–1.14; Figure 6 and Appendix A). The reduction in bleeding interests both cerebral bleeding (Figure 7A and Appendix A) and gastrointestinal bleeding (Figure 7B and Appendix A).

Therapy with DOACs modifies the ischemic-hemorrhagic risk profile of cardio-oncological patients. The reduction in major bleeding and the even more significant decrease in stroke rates determine a net clinical benefit, quantifiable in terms of the number needed to treat (NNT). For every 100 theoretical patients receiving DOACs, in fact, there is a reduction of a stroke event for every 21 patients. Similarly, NNT becomes 45 if the major bleeding covariate is considered. NNT to reduce the stroke “or” major bleeding event is only 14 demonstrating the extreme effectiveness of this therapeutic strategy (Figure 8).

The reduction in major and minor bleeding appears more significant as the percentage of females in the studies increases (Appendix A). The use of antiplatelet agents neutralizes the beneficial effect on the reduction of minor bleeding events but has no relevance to major bleeding events. Patients with higher CHA2DS2-VASc and HAS-BLED scores are those who benefit most from the reduction in cerebral and gastrointestinal bleeding events (Appendix A).

## 10. Discussion

In this comprehensive metanalysis of 11 studies [9,10,11,13,14,15,16,17,18,19,20] (incl. 266,865 patients) evaluating the safety and effectiveness of oral anticoagulants in cancer patients with AF, therapy with DOACs was significantly associated with greater stroke prevention and reduced bleeding than treatment with Warfarin.

The classic ischemic and hemorrhagic stratification scores (HAS-BLED and CHA2DS2-VASC) did not correlate with events in cancer patients. Our analysis shows al-so that patients at higher risk showed greater higher-risk-benefits in bleeding reduction, especially females and high-profile patients. This seems to be epically concerning also considering that recent reports have shown that even in high-risk patients (CHA2DS2-VASC > 2) where anticoagulation therapy is routinely recommended for life, about 44.3% of patients did not receive appropriate anticoagulation treatments [21,22]. Although international guidelines recommend DOACs over VKAs as the preferred anti-coagulant strategy in eligible AF patients, the claim cannot be automatically extended to oncologic patients because of their underrepresentation in randomized clinical trials (RCTs) [23]. The lack of use of anticoagulation may also be due to concerns from physicians mainly linked to elevated bleeding risk in cancer patients, lack of dedicated risk scores the need for dose modification, and drugs to-drugs interaction [8].

The management of thrombosis and bleeding in cancer patients requires careful consideration of the competing risks in each individual patient, even more, when atrial fibrillation is present as well: questions on therapeutic strategy remain in fact unanswered and inadequately studied. Despite better stratification and therapy duration, the increasing life expectancy of these cardio-oncological patients requires safe and effective medium-term algorithms.

Warfarin, the historic cornerstone of anticoagulant therapy, displays a narrow therapeutic window so that multiple-dose adjustments are often required to match the correct dosage with the optimal levels of coagulation. As Warfarin represents the leading cause of emergency room visits and hospitalizations for an adverse drug reaction [24], underuse in a lesser-studied population has been perpetuated, despite clinical indications.

In contrast, to VKAs, DOACs exhibit more predictable pharmacodynamic and pharmacokinetic properties, as they do not require routine coagulation monitoring, and show a relatively lower potential for drug interactions [25]. Antidotes for DOACs have also recently become available, allowing some reversibility of anticoagulation in life-threatening bleeding [26].

Despite an intuitive clinical benefit of DOACs with respect to VKAs, the metanalysis results display an unpredictable greater-than-expected reduction of ischemic-hemorrhagic events than those of non-oncological patients. Therefore, despite the heterogeneity of studies and a few RCTs, DOACs should be preferred to vitamin K antagonists. However, it must be underlined how the benefits in terms of stroke reduction are offset by a relatively high risk of bleeding even with DOACs.

As shown in Figure 8, results should be interpreted in terms of “net clinical benefit”, which is the balance between stroke reduction and increased bleeding associated with treatment. In this perspective, the number needed to treat (NNT), and the number needed to harm (NNH) aid in weighing the pros and cons of treatment.

From the data of our metanalysis, favoring DOACs over Warfarin in AF cancer patients leads to a reduction of one cerebral ischemic event every 21 patients alongside the risk of causing a major bleeding event in 1 out of 45 patients treated, still lower than those occurring in patients receiving Warfarin. However, considering the combined benefit of reducing an ischemic or major bleeding event, the NNT value is 14.

As a matter of fact, in AF cancer patients the risk of stroke is more potential, whereas bleeding risk is augmented. The available guidelines are precautionary and insist on risk estimation with an individualized approach [8].

Specifically, the ISTH guidelines recommend an “individualized anticoagulation” in favor of DOACs, provided there are no significant drug-drug interactions. DOACs should instead be limited to patients with luminal gastrointestinal cancers or gastrointestinal mucosal alterations [27].

EHR guidelines recommend interdisciplinary teamwork to address thromboembolic and hemorrhagic risk in cancer patients balancing AF, cancer-related, and treatment-related risk factors [25].

Furthermore, ESC guidelines recommend individualized decisions on anticoagulation, considering other co-morbidities, bleeding risks, and patient values and preferences as well. The role of DOACs in this patient group remains to be clarified; however, large DOAC trials suggest their safety [28].

Definitive conclusions cannot be drawn in the absence of strong evidence; however, it is reasonable to focus on DOACs in AF cancer patients based on the founding principle “primum non nocere”. Dedicated studies and new risk scores that include the “cancer vari-able” in risk prediction will be necessary. From this point of view, the cardio-oncological scientific community must begin to extend and unite the cultural horizons of two different specialties which, however, end up treating the same patient.

Future studies will necessarily have to consider the new interventional options such as those of the pharmacological and percutaneous treatment of stroke, the pharmacological and interventional treatment of major bleeding, and cardiological treatments modulating the ischemic risk such as trans-catheter ablation and percutaneous closure of the left atrial appendage.

In those patients with high thromboembolic risk (CHADS2-VASc > 2) but concomitant contraindications to anticoagulation, left atrium appendage closure (LAAO) could be considered an indication [29].

The oncological population could theoretically be ideal for this type of treatment. Left atrial appendage closure is suggested by the current European consensus statement [30] there is still a dramatic lack of evidence; only one study at the moment is specifically focused on this element of our knowledge [31].

In 2022, the first joint ESC guidelines of the oncology and cardiology societies were published.

Cancer therapy-related cardiovascular toxicity risk is a dynamic variable and the risk changes throughout the pathway of care. Atrial fibrillation is also dynamic and its cardioembolic risk changes over time, influenced by the progression of cardiovascular aging and risk factors.

All types of cancer show an increased risk of AF compared with the control group, but the risk of AF depends on the cancer type and stage. AF during cancer treatment may be caused by a specific therapy or interaction with a pre-existing substrate in older patients with cancer.

The anticoagulation strategy is defined as a “complex issue” since NOAC, Warfarin and LMWH have limited Class IIa evidence and “should be considered” in cancer patients (weight of evidence/opinion is in favor of usefulness/efficacy). The evidence regarding percutaneous closure of the left atrial appendage is only IIB and this possibility “may be considered” in selected cancer patients (usefulness/efficacy is less well established by evidence/opinion) [32,33].

These guidelines confirm the limits of the HAS-BLED and CHA2DS2-VASc Score which, as per our analyses, do not correlate much with the ischemic-hemorrhagic risk. However, long-term anticoagulation is recommended in adult patients with CHA2DS2-VASc score ≥2 in men or ≥3 in women and the clinical pattern of AF (paroxysmal, persistent, long-standing persistent, permanent, post-operative) should not influence the indication of thromboprophylaxis.

The role of NOACs remains controversial (IIA) but is supported by secondary analyses of trials that suggest better safety and at least similar effectiveness of the NOAC when compared with VKA in patients with AF and active cancer. It is reiterated that NOAC use in cancer is limited by drug–drug interactions, severe renal dysfunction, increased risk of bleeding in patients with unoperated or residual GI or genitourinary (GU) malignancies, or impaired GI absorption.

Despite the limited evidence, the cardio-oncological ESC guidelines emphasize the importance of a constant periodic reassessment of the ischemic-hemorrhagic risk (evidence IC) and suggest the use of NOACs in preference to LMWH and VKA in patients without a high bleeding risk, significant drug–drug interactions, or severe renal dysfunction. Inter-disciplinary teamwork between different specialties will be deemed necessary for cancer care practices by applying an individual approach inspired by the best evidence-based clinical practice.

## 11. Conclusions

The present meta-analysis suggests a better safety efficacy profile of DOACs in AF cancer patients. The significant benefit in terms of stroke reduction, albeit limited by the bleeding diathesis, appears advantageous. Therefore, it would be reasonable to suggest DOACs as a first choice for cancer patients. However, further studies will be needed to understand which patients benefit least from this type of anticoagulation strategy.

## Figures and Tables

**Figure 1 cancers-15-02574-f001:**
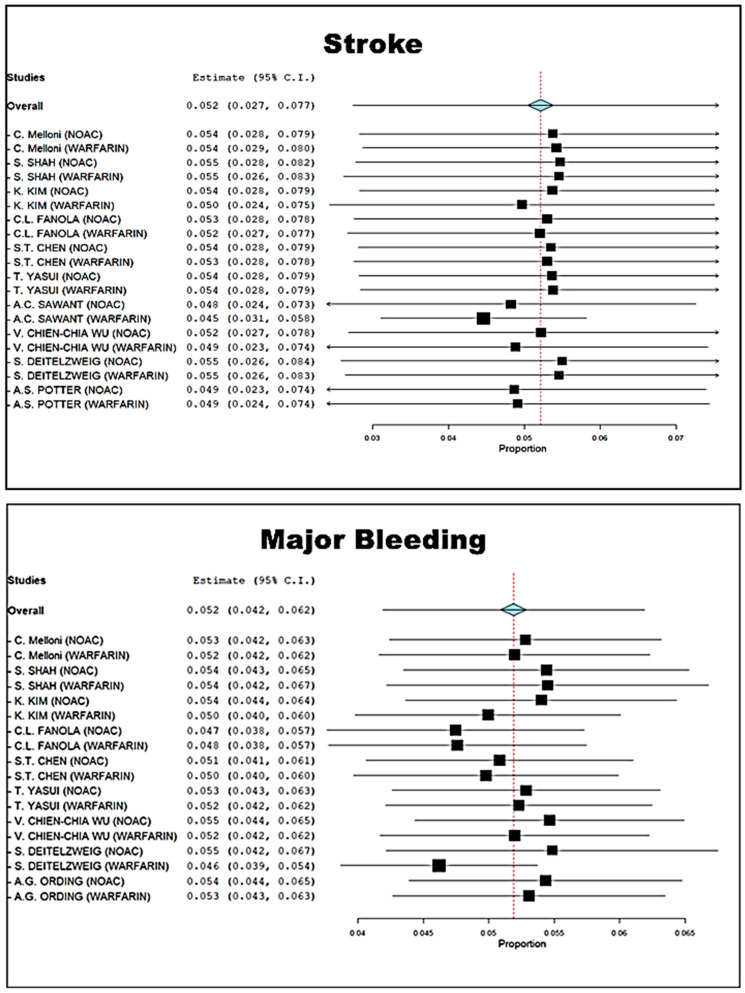
Cumulative meta-analysis of stroke and major bleeding of the included studies.

**Figure 2 cancers-15-02574-f002:**
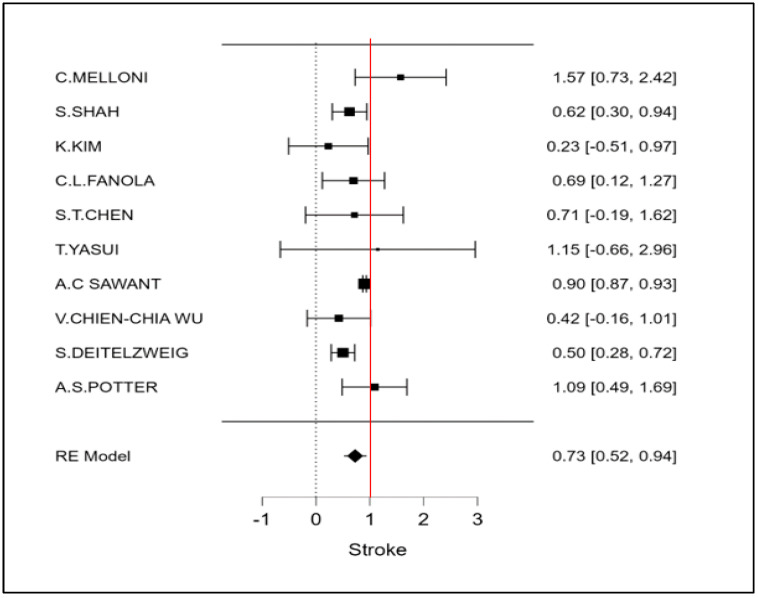
Meta-analysis of ischemic stroke.

**Figure 3 cancers-15-02574-f003:**
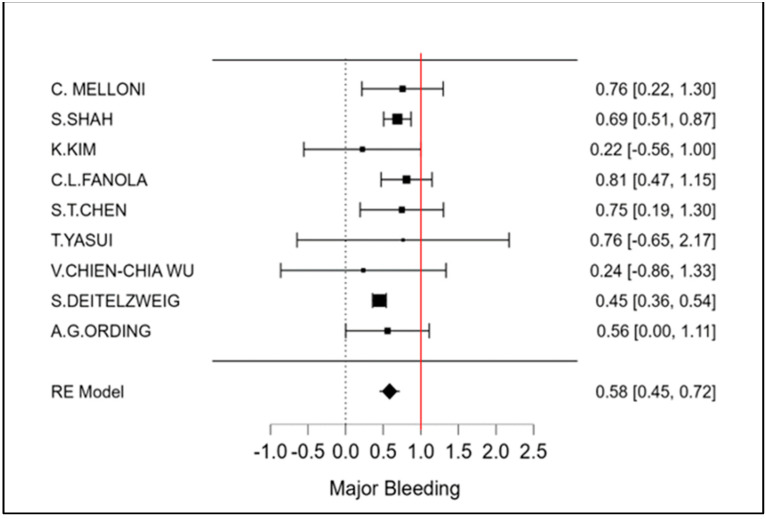
Meta-analysis of Major Bleeding.

**Figure 4 cancers-15-02574-f004:**
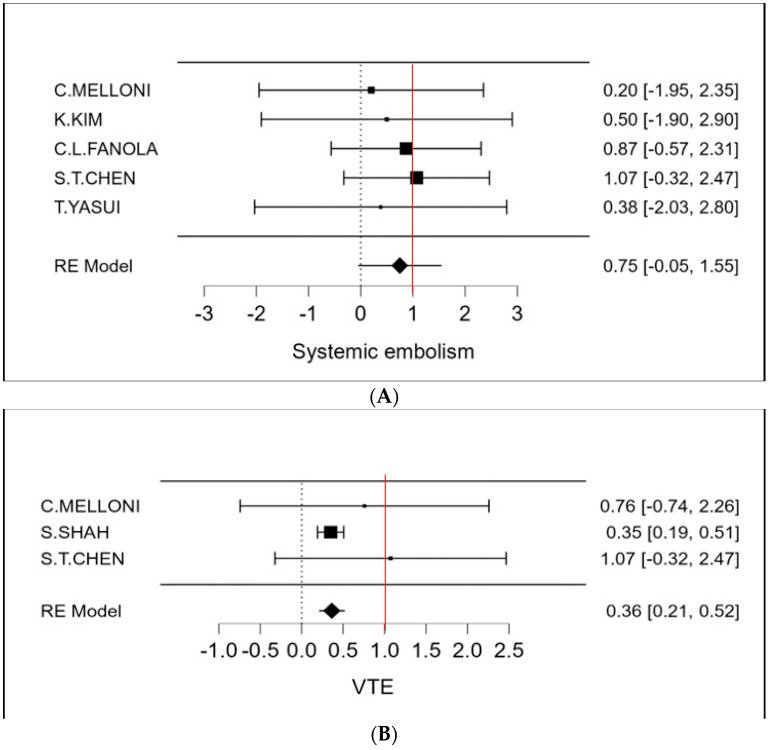
(**A**) Meta-analysis of systemic embolism; (**B**) Meta-analysis of VTE.

**Figure 5 cancers-15-02574-f005:**
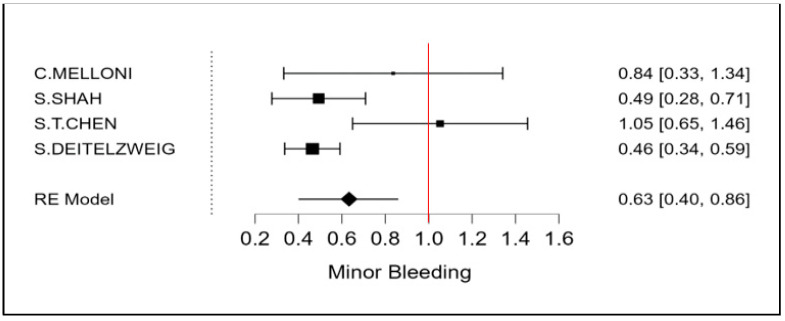
Meta-analysis of Minor Bleeding.

**Figure 6 cancers-15-02574-f006:**
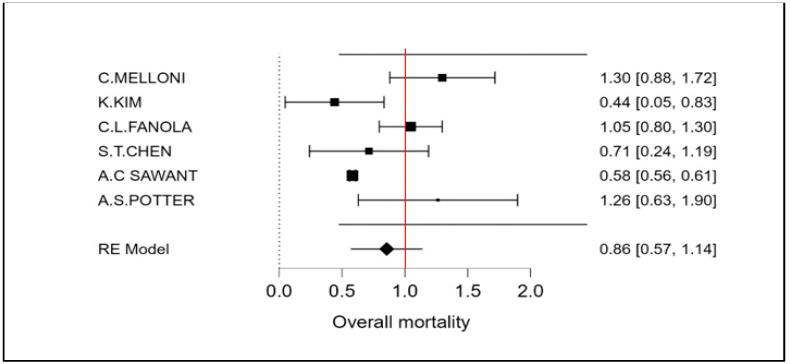
Meta-analysis of overall mortality.

**Figure 7 cancers-15-02574-f007:**
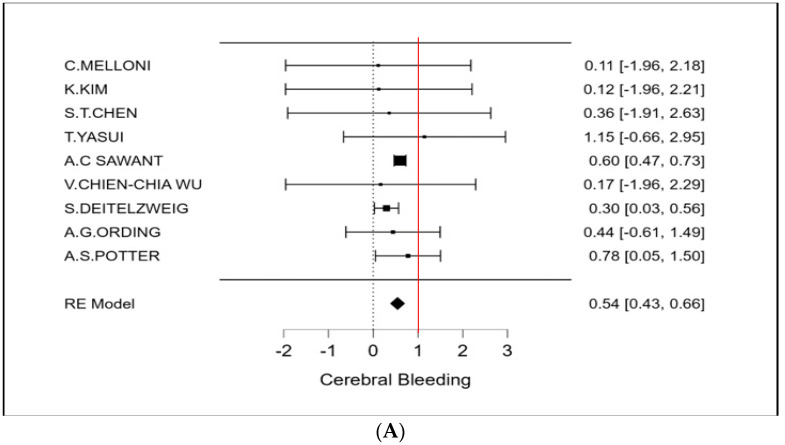
(**A**) Meta-analysis of Cerebral Bleeding; (**B**) Meta-analysis of GI Bleeding.

**Figure 8 cancers-15-02574-f008:**
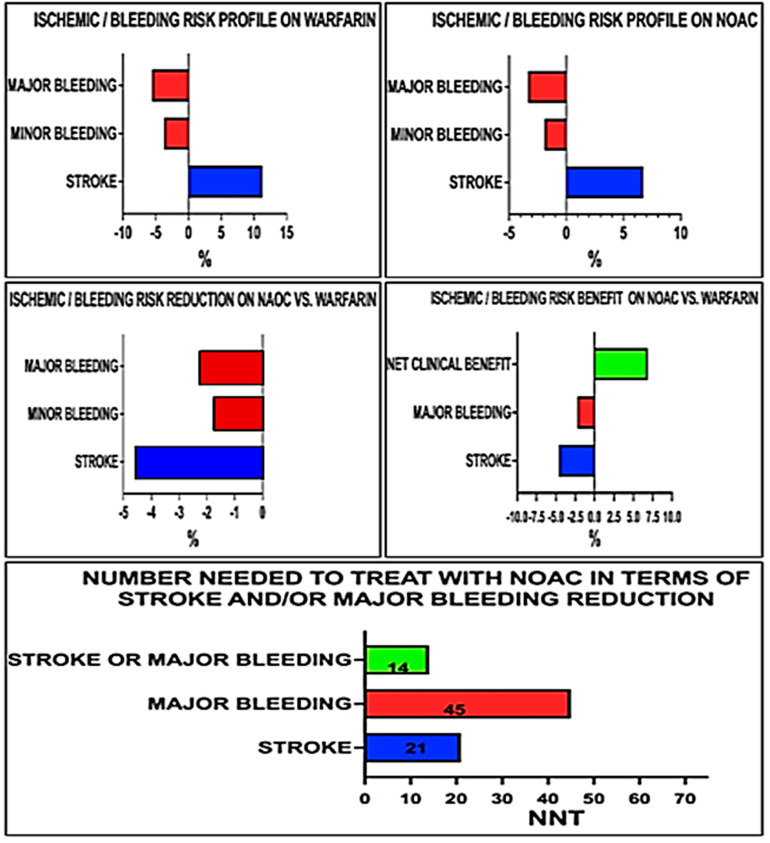
Clinical benefit of NOAC in terms of stroke/Major Bleeding reduction.

**Table 1 cancers-15-02574-t001:** Clinical profile of patients included in the meta-analysis.

Study	Mean Age	Female Sex %	CHA2DS2-VASc	Has-Bled	Apixaban	Dabigatran	Rivaroxaban	Edoxaban	Active Cancer	Follow-Up (Years)
Melloni	75.00	32.00	3.80	2.2	100.00	-	-	-	12.70%	1.80
Shah	74.70	40.00	4.40	-	17.70	36.00	46.00	-	100%	4.00
Kim	70.80	45.80	3.60	1.90	35.60	36.10	36.10	-	100%	1.70
Fanola	75.00	31.10	2.80	2.7	-	-	-	100.00	100%	2.80
Chen	77.00	34.00	3.50	2.9	-	-	100.00	-	7.80%	1.90
Yashui	72.00	14.00	3.10	2	36.20	19.70	34.60	9.40	100%	1.00
Sawant	76.00	20.00	-	-	26.1	41.2	32.7	-	100%	1.00
Chia	69.90	38.00	4.10	3.30	-	-	-	-	100%	1.50
Deitelzweig	77.00	39.00	4.00	3.50	24.00	7.00	31.00	-	100%	2.00
Ording	78.25	41.20	3.75	2.00	40.90	22.40	35.80	0.90	43%	1.00
Potter	71.50	71.00	3.40	1.90	57.20	6.40	36.00	0.40	100%	5.00

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
