# Peer review of "Systematic Review and Meta-Analysis of Oral Anticoagulant Therapy in Atrial Fibrillation Cancer Patients"

_cancers, 2023, doi:10.3390/cancers15092574_

Round 1

Reviewer 1 Report

1. the title of the paper is : Metanalysis of Oral Anticoagulant Therapy in Atrial 2 Fibrillation Cancer Patients and in the Methods section the autors state: This metanalysis (11 studies (incl. 266,865 patients) aims at evaluating the 53 ischemic-hemorrhagic risk profile of oncologic patients with AF treated with oral anticoagulants 54 (vitamin K antagonists vs direct oral anticoagulants). Please change the title or the aim of the paper, so they are more consistent.

  • 2. In abstract section, line 50 autors state that: These two pathologies share an increased thrombotic and bleeding risk. - can be changed ? maybe use "conditions"

3. Introduction: it will be interesting for the reader to indicate what percentage of patients with atrial fibrillation are diagnosed with cancer, oncological diseases. Why oncology patients are at higher risk of bleeding? maybe the autors should add some information eg.he the relationship between the cancer process and blood clotting is actually reciprocal - cancer induces a hypercoagulable state,which remains a major risk factor for venous thromboembolism ...

4. The autors whrite: As cancer patients were not included in large cardiovascular studies, the potential 94 benefit of direct oral anticoagulants (DOACs) appears to be reasonable but not 95 supported by evidence at present times [9,10,11]. 

Please check: Larsen TB, Nielsen PB, Skjøth F, et al. Non-vitamin K antagonist oral anticoagulants and the treatment of venous thromboembolism in cancer patients: a semi systematic review and meta-analysis of safety and efficacy outcomes., Bauersachs R, Berkowitz SD, Brenner B, et al. EINSTEIN Investigators, EINSTEIN Investigators. Oral rivaroxaban for symptomatic venous thromboembolism. N Engl J Med. 2010; 363(26): 2499–2510, Büller HR, Prins MH, Lensin AWA, et al. EINSTEIN–PE Investigators. Oral rivaroxaban for the treatment of symptomatic pulmonary embolism. N Engl J Med. 2012; 366(14): 1287–1297 and other 

Results

1. The authors in Table 1 showed the total score for the has bleed and Chadsvas scales - is it possible to show separate components of the scales? 

2. please use the correct scale nomenclature CHA2DS2-VASc in the text 

3. figure 1 is unreadable

Overall, The authors, in their conclusion, point out the safety of using DOACs in cancer patients, however are there any data comparing the use of DOACs or VKAs in this group? It is well known that the use of DOACs is safer and is associated with a lower risk of acidemia and is more convenient for patients 

Are patients diagnosed with AF and cancer also more compliant with therapy?

Author Response

Replies to Reviewer 1

1) We agree, we changed the part of the scope of the study.

2) Ok, edited pathologies with conditions.

3) We have enriched the introduction by trying to address the issues. There are many pathophysiological explanations. Being a somewhat orphaned research area, the purpose of our meta-analysis is to try to humbly fill this gap of knowledge.

4) We have added the references, a very important article which however is limited to thrombo-embolic events (venous thrombosis and pulmonary embolism). Our meta-analysis is focused on the prevention of systemic cardio-embolic events (mainly stroke) associated with atrial fibrillation. We have cited both references in the conclusions.

The request to show the separate Has-Bled and Cha2ds2-Vasc values ​​in Table 1 is pertinent because it is a source of potential bias. The raised issue is partially compensated by the existence of Figure 1 (reviewed in graphic quality as per the reviewer's suggestion) and by Figure 2 of the supplementary materials where the average value of the scores in patients with NOAC and Warfarin is reported. There was no statistically significant difference.

The nomenclature of the scores has been revised throughout the text.

A more readable Figure 1 in png format has been sent.

There are no dedicated data in the literature on the compliance of AF cancer patients, an interesting starting point for future research.

Our conclusions, as reported in the Conclusions, in favor of NOACs are supported by data from our meta-analysis. The reduction in cardiovascular events and bleeding is significant. There is no impact on overall mortality.

Reviewer 2 Report

In the present manuscript by Alberto Cereda et al. titled “Metanalysis of Oral Anticoagulant Therapy in Atrial 2 Fibrillation Cancer Patients”, the authors aim to compare the benefit of direct oral anticoagulant (DOACs) in terms of stroke, major and minor bleeding and thromboembolic events. This manuscript showed meta-analysis gathering the use of DOACs in atrial fibrillation cancer patients compared to warfarin.  This analysis include 11 studies evaluating the ischemic and hemorrhagic risk of oncologic patients with AF under anticoagulation therapy. In conclusion, anticoagulation with DOACs provides higher safety profiles than vitamin K antagonist.

The manuscript is very well written and provides beneficial insight of anticoagulation therapy. I believe that the manuscript could be accepted with revision. I have provided some suggestions on ways to improve the manuscript, which are provided in the comments below.

Comments regarding revision:

-          The language is understandable, however some grammatic errors are present. The authors should correct them or use an English language editor to help them.

-          The introduction is precise and understandable, but efficacy and risk of using DOAC were estimated by previous papers. The authors could mention more clearly already known results in patients without cancer.

-          Table 1 shows clinical profiles, if available type of atrial fibrillation (paroxysmal or non-paroxysmal) should depict.

-          The authors discussed about ischemic and haemorrhagic stratification scores (HAS-BLED and CHA2DS2-VASc), the result about risk scores showed only in the supplement. The figure 2 and 3 in the supplement should be showed in the main manuscript.

Author Response

Replies to reviewer 2

We've adjusted and revised the text styling.

Data divided by types of atrial fibrillation are available in only a few studies. The most recent guidelines say they don't distinguish between the various forms of atrial fibrillation. Also, in my opinion the burden of AF is important but it has been minimized by the cardiological guidelines.

The article has many tables, and we have also added the graphical abstract. This observation was somewhat prompted by reviewer 1. We emphasized in the caption of Table 1 that additional information is present in the supplementary materials.

Round 2

Reviewer 1 Report

Please check your entire paper for missing literature citations, f.eg:

This seems to be epically concerning also considering that recent reports 261 have shown that even in high riskhigh-risk patients (CHA2DS2-VASC>2) where 262 anticoagulation therapy is routinely recommended for life, about 44,3% of patients did 263 not receive appropriate anticoagulation treatments

the authors state that recent reports - which reports?

Author Response

We have added two more references, 5 since the first revision for a total of 34 bibliographic citations

We checked the reference in question

The report in question corresponds to the reference 22 of Fradley et Al (DOI: 10.1016/j.jaccao.2020.09.008)

The paper was published in 2020 (we cannot define it as not recent) and is still an important journal in the cardio-oncological field

Quoting the text

"Nearly one-half of patients with cancer, the majority with normal platelet counts, had an elevated risk for stroke but did not receive anticoagulation. In addition to known predictors, current chemotherapy use was independently associated with lower odds of AC use. This study highlights the need to improve the application of AF treatment algorithms to cancer populations"

This confirms the contemporaneity and importance of our review

These are reasonable numbers in clinical practice, let us think of a recurrence of atrial fibrillation after surgery in a patient who has to start adjuvant chemotherapy. Anticoagulation is often omitted and replaced by underdosed prophylactic LMWH